# ToolkenGPT: Augmenting Frozen Language Models with Massive Tools via Tool Embeddings

**Shibo Hao**[1], **Tianyang Liu**[1], **Zhen Wang**[1,2], **Zhiting Hu**[1]
[1]UC San Diego, [2]Mohamed bin Zayed University of Artificial Intelligence
{s5hao, til040, zhw085, zhh019}@ucsd.edu

## Abstract

Augmenting large language models (LLMs) with external tools has emerged as a promising approach to solving complex problems. However, traditional methods, which fine-tune LLMs with tool demonstration data, can be both costly and restricted to a predefined set of tools. Recent in-context learning paradigm alleviates these issues, but the limited context length only allows for a few shots of demonstrations, leading to suboptimal understandings of the tools. Moreover, when there are numerous tools to choose from, in-context learning could completely fail to work. In this paper, we propose an alternative approach, **ToolkenGPT**, which combines the benefits of both sides. Our approach represents each tool as a token ("*toolken*") and learns an embedding for it, enabling tool calls in the same way as generating a regular word token. Once a toolken is triggered, the LLM is prompted to complete arguments for the tool to execute. ToolkenGPT offers the flexibility to plug in an arbitrary number of tools by expanding the set of toolkens on the fly. In addition, it improves tool use by allowing extensive demonstration data for learning the toolken embeddings. In diverse domains, including numerical reasoning, knowledge-based question answering, and embodied plan generation, our approach effectively augments LLMs with tools and substantially outperforms various latest baselines. ToolkenGPT demonstrates the promising ability to use relevant tools from a large tool set in complex scenarios.[1]

## 1 Introduction

Large Language Models (LLMs) [5, 9, 62, 47] have established themselves as powerful tools for diverse real-world applications, ranging from writing assistance to automated customer support [2, 6, 14]. As these models continue to evolve, there is a growing interest in their potential to interact with the real world and enhance their functionality through integration with other tools, such as the calculator, databases, etc [50, 61, 56, 53]. The capability of these models to master and control a wide array of tools not only serves as an indicator of their intelligence, but also signals a promising path to overcome some of their fundamental weaknesses. These include updating the latest world knowledge [46], reducing their hallucinations [55, 58], and executing symbolic operations [12, 17, 48], etc. However, the rapid emergence of new tools, such as advanced software libraries, novel APIs, or domain-specific utilities [39, 36, 29], introduces additional richness and complexity to the task of tool learning for LLMs. This continuous evolution accentuates the importance of empowering LLMs with the ability to adapt and master massive new tools swiftly.

Recent advancements in LLMs have witnessed two primary lines of research approaches for tool integration with LLMs [45, 68, 53] (Table 1). The first paradigm involves fine-tuning LLMs to learn specific tools [50]. For example, there are enormous efforts to integrate the retrieval tool into

---

[1]Code is available at `https://github.com/Ber666/ToolkenGPT`

37th Conference on Neural Information Processing Systems (NeurIPS 2023).

Table 1: Comparison of different tool learning paradigms. Plug-&-Play means the LLMs can be equipped and unequipped with a tool flexibly. Note that it doesn't indicate zero-shot tool learning.

| Tool Learning Paradigms | Frozen LMs | Massive Tools | Plug-&-Play | Ability to Use Extensive Data |
|---|---|---|---|---|
| Fine-tuning [e.g., 56, 50] | ✗ | ✗ | ✗ | ✓ |
| In-context learning [e.g., 69, 53, 7] | ✓ | ✗ | ✓ | ✗ |
| ToolkenGPT (**Ours**) | ✓ | ✓ | ✓ | ✓ |

LLMs [18, 35, 58, 3] and the recent Toolformer [56] fine-tuned GPT-J to learn five tools. While this method could yield promising results, it is computationally expensive and lacks the adaptability to new tools. The second approach relies on in-context learning [69, 49, 53], where LLMs learn how to use the tool through in-context demonstrations provided in the prompt. This method allows LLMs to handle newly introduced tools and drives successful applications like LangChain [7] and ChatGPT Plugin [2]. However, in-context learning comes with its own unique limitations. Specifically, it struggles with the inherent limitation of context length, making it impossible to demonstrate massive tools in the context. Also, mastering new tools simply via few-shot examples could be challenging. For example, even the latest models like GPT-4 face difficulties when handling unusual tools [6].

In this paper, we introduce ToolkenGPT, an alternative solution that enables LLMs to master massive tools without the need for any LLM fine-tuning, while still allowing for quick adaptation to new tools. The key idea of ToolkenGPT is to represent each tool as a new token ("*toolken*") to augment the vocabulary. Specifically, each tool is associated with an embedding inserted into the LLM head like a regular word token embedding. During generation, once a toolken is predicted, the LLM temporarily switches into a special mode (through prompting) to produce input arguments for the tool to execute, and inject the outputs back into the generation (see Figure 1). This approach offers an efficient way for LLMs to master tools by only learning the lightweight toolken embeddings. Consequently, ToolkenGPT combines the strengths of both fine-tuning and in-context learning paradigms while avoiding their limitations (Table 1): Compared to in-context learning that can only accommodate a small number of tools and few-shot demonstrations, ToolkenGPT allows massive tools (by simply inserting respective toolkens in the vocabulary) and can use extensive demonstration data for learning toolken embeddings; In contrast to LLM fine-tuning, the tool embeddings not only requires minimal training cost, but also provide a convenient means for plugging in arbitrary new tools on the fly by expanding the toolken vocabulary.

We demonstrate the flexibility and effectiveness of our ToolkenGPT in leveraging numerous external tools for solving a diverse set of problems, spanning from numerical reasoning to knowledge-based question answering and embodied plan generation. In complex numerical reasoning problems that involve a number of mathematical tools (numerical operations such as finding *greatest common divisor*), we show that ToolkenGPT can effectively utilize these tools during the reasoning process, which outperforms some of latest popular approaches, such as Chain-of-Thought [65] and ReAct [69]. For knowledge-based question answering, ToolkenGPT accommodates a substantial number of relation APIs (over 200) from the knowledge base, thereby facilitating factual predictions. Furthermore, we apply our framework to task planning for embodied agents, where an agent interacts with an environment using tools, namely the actions and objects. The findings illustrate that our method offers better grounding by learning toolken embeddings for 58 grounded actions and objects than previous in-context learning and specialized decoding methods.

## 2 Related Works

**Fine-tuning LLMs to use tools.** Early research relied heavily on fine-tuning to augment LMs with tools. In these works, LMs were mostly fine-tuned to use one or a few tools in a specific domain. For example, the retriever has been a crucial tool for augmenting LLMs with external knowledge sources [70]. The prominent works in this line include REALM [18], RAG [35], and RETRO [3]. More recently, WebGPT [46] fine-tuned GPT-3 on human web search behaviors to learn how to use the web browser. With the advancements in LLMs, there has also been growing interest in tuning

---
[2]https://openai.com/blog/chatgpt-plugins

these models on a collection of general tools, including the QA model, calculator, translator, etc. Example works include TALM [50] and Toolformer [56]. However, LLM fine-tuning is costly and these tuned LLMs struggle to generalize to emergent or updated tools. ToolkenGPT learns lightweight toolken embeddings for new tools, without any gradient calculation for the parameters of LLMs. This enables efficient adaption to new tools and maintains a minimal GPU memory overhead for training toolken embeddings, at a cost similar to LLM inference.

**In-context learning for tools.** LLMs exhibit a strong in-context learning ability [5], which becomes a prevalent method to use tools by showing tool descriptions and demonstrations in context [45, 53]. Building on this idea, reasoning chains can be incorporated to tackle more complex problems [69, 32, 49]. This paradigm has given rise to popular industry products such as ChatGPT plugins and Langchain [7], along with many successful applications in important research topics. For instance, a code interpreter can effectively address the LLM's shortcomings in symbolic operations [8, 17, 21, 44, 67, 40]. Furthermore, by calling "tools" that have an effect on the virtual or physical world, the LLM is capable of guiding embodied agents to accomplish various household tasks [25, 4, 26, 59, 27]. Recent attempts to utilize LLMs as a controller to coordinate multiple neural models also achieve promising progress in multimodal reasoning tasks [57, 43]. Nevertheless, all methods based on in-context learning suffer from inferior performance in complex scenarios, where the tools are unfamiliar or numerous. One concurrent work, Li et al. [36] propose to retrieve the tools based on the text embedding of their documents, which may mitigate that issue. However, ToolkenGPT is fundamentally different from their method, in that the toolken embeddings can encode the implicit semantics of tools from extensive demonstrations, which can never be inferred from the surface text (A concrete example is shown in Figure 3). Also, note that ToolkenGPT is compatible with the recent advanced prompting techniques, e.g., Chain-of-Thought (CoT) [65], to improve the LLMs performance further.

**Efficient tuning of large language models.** Adapting pre-trained frozen LLMs efficiently to new tasks is an active research area, leading to a surge of interest in parameter-efficient fine-tuning (PEFT) methods [31, 38, 11, 42, 41]. The idea is to only fine-tune a small subset of parameters of the LLM while freezing most of its parameters, which bears similarity to our toolken embedding method. Which part of parameters to tune is the key to PEFT methods; for instance, Adapters [22] insert trainable layers, BitFit [71] tunes the bias parameters, prompt tuning [34, 64] appends parameters to the input embedding layer, and LoRA [23] learns low-rank matrices within specific dense layers, etc. However, existing PEFT methods have not proven suitable for efficient tool learning, and utilizing these methods on tool demonstrations may not efficiently capture the desired tool knowledge as ToolkenGPT does. To the best of our knowledge, we are the first to explore efficient tuning methods for predicting tools as tokens for tool learning of massive tools.

# 3 ToolkenGPT for Mastering Massive Tools

In this section, we present ToolkenGPT, which enables LLMs to learn and use massive tools for complex problem-solving without the need for heavily fine-tuning the LLM. We begin by introducing the background and notations of language modeling for tool use. Typically, LLMs model the probability of a sequence of word tokens $s = (t_1, t_2, ..., t_n)$ as $P(s) = \sum_i^n P(t_i \mid t_{<i})$, where each word token comes from the vocabulary of the LLM, i.e. $t_i \in \mathcal{V}$ and $t_{<i}$ denotes the partial word token sequence before $i$-th step. In practice, the user often sets the prefix of a sequence (referred to as the prompt) to steer LLMs to generate desired contents, e.g., answering a question. Taking a step deeper, the distribution of the next token is predicted as $P(t_i|t_{<i}) = \text{softmax}(W_\nu \cdot h_{i-1})$, where $h_{i-1} \in \mathbb{R}^d$ is the last hidden state of the current context and $W_\nu \in \mathbb{R}^{|\mathcal{V}| \times d}$ is the embedding matrix for word tokens (also known as language model head).

Given a set of useful tools $\mathcal{T} = \{\tau_1, \tau_2, ...\}$, our goal is to enable LLMs to call a subset of these tools for solving the complex problem. Our flexible formulation allows tools to play a role by either returning some results that can help LLMs with text generation (e.g. calculation) or affecting the real-world environment (e.g. robot action). To call a tool during generation, the LLM first needs to select a tool and then input the arguments. In the running examples shown in Figure1, during the answer generation process ("reasoning mode"), a math operator `square` is selected as the tool, and an operand `16` is generated as the argument in the "tool mode". Once the external tool receives the call, it executes the tool and returns the result `256`, back to the "reasoning mode".

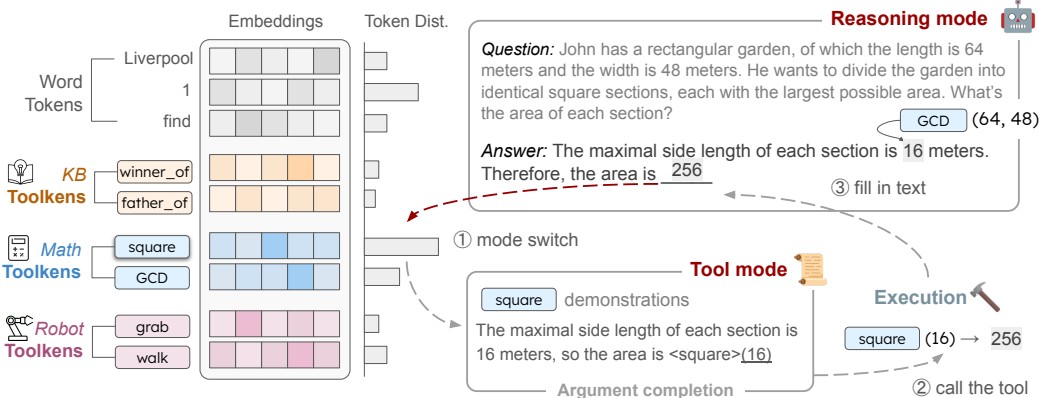

Figure 1: Overview of ToolkenGPT framework. Toolken embeddings are appended to the language model head like regular word tokens. In the "reasoning mode" for solving the problem, the LLM generates text as usual, except that any plugged-in toolkens are also considered for the next token generation. Once a toolken is predicted, (1) the LLM switch to the "tool mode", which provides a few demonstrations of the same tool to complete the arguments. Then, (2) the tool call is executed, and (3) the result is sent back to the text to continue the reasoning mode until the final answer is generated.

## 3.1 Framework Overview

The core idea of ToolkenGPT is explicitly formulating tools as tokens (called "*toolkens*"). Each toolken is parameterized as a *toolken embedding* vector, and we denote a set of toolken embeddings as a matrix, i.e. $W_\tau \in \mathbb{R}^{|\mathcal{T}| \times d}$. Assuming we have trained toolken embeddings (to be described in Section 3.2), we first give an overview of our framework by introducing how it works in inference. As shown in Figure 1, the LLM is in the reasoning mode by default, generating the next token. Our framework allows the LLM to consider word tokens and toolkens uniformly. Specifically, the tool embedding matrix is concatenated with $W_\nu$. Therefore, the LLM predicts the next token with the probability as follows:

$$P(t_i|t_{<i}) = \text{softmax}([W_\nu; W_\tau] \cdot h_{i-1}) \tag{1}$$

where the next token can be either a word token or a toolken, i.e. $t_i \in \mathcal{V} \cup \mathcal{T}$, and $[;]$ is the concatenation operation. As we can see, our formulation of tools as toolken embeddings naturally allows for the fast adaption of new tools by expanding the toolken embedding matrix easily.

To execute a tool, the LLM switches into the "tool mode" once its toolken is predicted as the next token (as shown in the "mode switch" in Figure 1), which aims to generate the arguments for the tool. Specifically, the LLM pauses the generation and appends the current generated context to another prompt. The prompt in tool mode consists of in-context demonstrations for the predicted tool, showing how to generate the tool arguments by quoting the tool calls in a special syntax of `[tool](arguments)`. Then the LLM can follow the pattern in demonstrations to complete the arguments of the current tool call. Contrasting previous methods [69, 53] that fully rely on in-context learning for tool learning, our framework only leaves the easy work of completing arguments to in-context learning. Besides, there would be abundant context space for extensive demonstrations of a single specified tool. This design shares similarities with the classic divide-and-conquer methods [33, 32, 13]. Finally, the arguments are sent to the specified tool for execution, and the returned value is sent back to the text in the reasoning mode.

## 3.2 Learning Toolken Embeddings

Our framework keeps the original LLM parameters frozen and introduces a minimal additional training overhead with the toolken embeddings, $W_\tau$. This embedding matrix contains the only parameters to optimize, but unlike other efficient LLM tuning methods, e.g., prompt tuning [34, 64] or prefix tuning [38], it does not require the gradients flowing through the major body of LLM parameters, leading to much stable and efficient training. Therefore, the tuning of toolken embeddings

maintains nearly the same GPU memory as LLM inference. Whenever a new tool is added, the toolken embedding can be conveniently expanded and then, subsequent training on tool demonstration data involving the new tool gradually refines its embedding. Moreover, unlike in-context learning methods that only digest a few examples as training signals, ToolkenGPT is capable of tuning toolken embeddings from massive demonstrations.

Drawing parallels to how infants learn a new tool through demonstrations from adults [15], in this paper, we primarily focus on learning toolken embeddings with tool demonstrations, which can be either in-domain training data or synthetic data generated by LLMs (see Section 4.1 and Section 4.2). We first describe the format of training data and the training objective and we use the same example from Figure 1 to showcase how it can be used for training. Specifically, *"the area is 256 square feet ..."* can be tokenized into a word token sequence $s$ = ("the", "area", "is", "2", "5", "6", "square", "feet", ...). To indicate when to predict the toolkens, we need a parallel sequence mixed with word tokens and toolkens, i.e. $s'$ = ("the", "area", "is", "[square]", "[N/A]", "[N/A]", "square", "feet", ...). The subsequence of ("2", "5", "6") in $s$ is where the returned tool results should fill in, and we choose the corresponding first token in $s'$ as the toolken for the tool call with the following tokens are filled with [N/A], indicating neglect in loss calculation. Thus, given a dataset composed of paired sequences $\mathcal{D} = \{(s, s')\}$, the training objective of ToolkenGPT is:

$$\mathcal{L}(W_\tau) = \sum_{(s,s')\in\mathcal{D}} \sum_{i=1}^{N} -\log P(t'_i|t_{<i}) \mathbb{1}_{t'_i \neq [\text{N/A}]} \tag{2}$$

where $P(t'_i|t_{<i})$ is defined in Eq.(1), and $\mathbb{1}_{t'_i \neq [\text{N/A}]}$ is the indicator function signaling we ignore the [N/A] tokens during the training. Thus, our training process is largely consistent with the inference in the reasoning mode. That is, to call a tool, the only job for the LLM is to predict a toolken at the beginning, and then the returned value will be filled back to the text. Here, [N/A] is introduced to skip the generation of the returned value of a tool call.

There are two primary ways to get the paired data. First, some datasets provide ground truth tool calls along with natural language sequences, e.g. the facts in KB supporting the answer to a question (Secion 4.2), or the calculation trace for solving a math problem (Section 4.1). To use the data for supervised learning, we preprocess them to get the paired data required for training as described in the above paragraph. Second, we explore synthesizing tool demonstrations with LLMs, sharing a similar idea to self-instruct [63]. An intuitive interpretation of this process is to distill the knowledge inside LLM to the new toolken embeddings. Specifically, we can prompt LLMs with the tool document and a few demonstrations with a special syntax indicating tool calling, e.g., *The capital of U.S. is <capital> ("U.S.")="Washington D.C."* Conditioned on that, the LLMs can generate some new use cases that utilizes the given tool and quote the tool call with the same syntax. We can then easily locate the tool calls and process the data into the paired data for training.

## 4 Experiments

In this section, we apply ToolkenGPT to three distinct applications characterized by meaningful tool-use scenarios: arithmetic tools for numerical reasoning, database APIs for knowledge-based question answering, and robot actions for embodied plan generation. We focus on how methods can accurately call the tools and how successfully they can solve the tasks. Our experiments show that ToolkenGPT can efficiently master massive tools while leveraging them to solve complex problems with improved performance, consistently better than advanced prompting techniques.

### 4.1 Numerical Reasoning

LLMs often struggle with mathematical tasks since the models are inherently designed for probabilistic estimation rather than symbolic operations. In this section, we aim to assess the tool-learning capabilities of ToolkenGPT, compared with in-context tool learning (e.g., ReAct [69]). We first demonstrate that ToolkenGPT consistently matches or outperforms the performance of in-context learning with the availability of four basic arithmetic functions ($+, -, \times, \div$). Moreover, to benchmark the tool-handling capability in more complex math problems, we include more available tools, i.e., an expanded (13) set of functions, and create a set of synthetic data. The results show that ToolkenGPT significantly outperforms baselines by training only on the synthetic data. Note that our focus is not to reach a state-of-the-art accuracy; Rather, the experiment is designed to evaluate the tool learning ability in the setting where certain tools are available.

Table 2: Results on the GSM8K-XL and FuncQA datasets. The numbers in parentheses indicate how many available tools are available. For GSM8K-XL and FuncQA$_{one}$ dataset, accuracy is evaluated based on an exact match (float numbers rounded to two decimals). In FuncQA$_{multi}$, we allow a margin of error of $0.1\%$ to account for potential errors at each step of multi-hop reasoning.

| Method | GSM8K-XL (4) | FuncQA (13) | |
| --- | --- | --- | --- |
| | | One-Hop | Multi-Hops |
| 0-shot ChatGPT | 0.17 | 0.55 | 0.09 |
| CoT [65] | 0.18 | 0.20 | 0.03 |
| ReAct [69] | 0.32 | 0.57 | 0.06 |
| ToolkenGPT (Ours) | **0.33** | **0.73** | **0.15** |

**Datasets.** To evaluate the tool-learning proficiency in numerical reasoning comprehensively, we curate two new test datasets: (1) **GSM8K-XL**, an enhanced version of the existing GSM8K [10] dataset. GSM8K is a dataset of linguistically diverse grade school math word problems, involving performing a sequence of calculations using 4 basic arithmetic operations $(+, -, \times, \div)$ to reach the final answer. In the original GSM8K dataset, the numbers for calculations are typically small, which might be less challenging for the recent powerful LLMs [10, 6]. So in the test set, we magnify the numbers to increase the computational difficulty for LLMs, which results in the GSM8K-XL dataset, featuring 568 test cases with much larger numbers. (2) **FuncQA** is a synthetic dataset we created to increase the complexity of math problems involving more arithmetic tools, which serves as a much more challenging benchmark to test the model's tool-learning capabilities. Specifically, This dataset requires at least 13 operators (e.g., power, sqrt, lcm) to solve, and it is challenging for both humans and LLMs to solve without an external calculator. Furthermore, FuncQA is categorized into two subsets: 68 one-hop questions (FuncQA$_{one}$) solvable with just one operation, and 60 multi-hop questions (FuncQA$_{multi}$) requiring a few reasoning steps.

To train the toolken embeddings used in GSM8K-XL, we preprocess the original training set of GSM8K which has the calculation annotation as described in Section 3.2. We get 6,054 examples, of which 1,000 were allocated for validation, and 5,054 for the training data. For the FuncQA dataset, we prompt ChatGPT to generate some one-hop QA patterns for each operator, and then randomly assign values to the patterns. This process yields 47 training data points and 3 validation data points for each operator, resulting in a total of 611 samples for training and 39 samples for validation.

**Comparison methods.** We train toolken embeddings for each available math operator as described in Section 3.2. During inference, we prompt the LLM with 4-shot Chain-of-Thought [65] examples to enhance the reasoning ability of LLMs. The following baselines are evaluated for comparison: (1) *0-shot CharGPT* is the straightforward method asking LLMs to answer a question. No examples will be provided in the context and tools are not available. We use ChatGPT as the base LLM in our experiment. This baseline measures the ability of the LLM to answer complex numerical reasoning problems with its own reasoning and calculation ability. (2) *Chain-of-thougts (CoT)* [65] is a more advanced prompting techniques. In this approach, a series of interconnected prompts are carefully crafted to guide the LLMs through a step-by-step reasoning process. The example reasoning chains are the same as the ones we used for ToolkenGPT, but no functions are available. (3) *ReAct* [69] combines reasoning and tools by prompting the LLMs to generate verbal reasoning traces and tool calls in an interleaved manner. Concretely, instead of just providing reasoning chains such as "... *The cost is 50\*3.2=160*", ReAct incorporates special syntax to call operators, e.g."... *The cost is 50\*3.2=<multiply>(50,3.2)=160*". Once the syntax is detected during inference, the tool would be called to calculate the result. We use the same reasoning chain examples as in both CoT and ToolkenGPT, with only slight differences in the tool calling syntax. LLaMA-33B [62] is used as the LLM for all settings other than zero-shot prompting. More experiment details are described in Appendix A.

**Result analysis.** Table 2 shows the performance of all the methods on the GSM8K-XL and FuncQA datasets. On the GSM8K-XL dataset, 0-shot ChatGPT and few-shot learning with CoT struggle to calculate large numbers without the help of tools, while ReAct and ToolkenGPT manage to increase accuracy consistently by a large margin. Generally, both methods can call the correct tools when necessary, as the toolset is comprised of only the four basic operators. However, for both FuncQA$_{one}$ and FuncQA$_{multi}$ datasets, learning to call applicable tools becomes challenging to ReAct as the

number of tools increases. In ReAct, though all the tools are listed at the beginning of the prompt, it is infeasible to include demonstrations of every tool in the limited context (In our experiment, we provide 4 examples including 5 tool demonstrations). As a result, ReAct is susceptible to missing tool calls, making wrong tool calls, and predicting wrong arguments, especially for the tools not demonstrated in context. ToolkenGPT outperforms all the baselines across both one-hop and multi-hop scenarios, showing superior tool learning ability when there are numerous tools. It is important to note that even though toolken embeddings are trained solely using *one-hop synthetic data*, and *without any CoT examples*, they still manage to enhance performance in multi-hop problem contexts and can be integrated effectively with CoT prompting. This implies a degree of generalization of toolken embeddings, which is a very desired property that lowers the requirements of in-domain training data.

## 4.2 Knowledge-based Question Answering

LLMs are known to often make factual errors and hallucinate [28, 73, 72, 1] because of their limited knowledge [20]. Equipping them with access to knowledge bases (KBs) has been a promising research direction to reduce their hallucinations [58]. We formulate the access to the KB as APIs querying the database [60, 16]. Thus, each relational query can be treated as a tool to which the input argument is a subject entity, and the output is the corresponding tail entity. An example tool call is "P1346(2005-06 FA CUP) → LIVERPOOL F.C." "P1346" is a relation identifier in Wikidata, representing the winner of a competition or similar event (referred to winner_of below for ease of reading). In this section, we show that ToolkenGPT can accurately query a large knowledge base of *up to 234 tools (relations)*. We further show that even *only with synthetic data* (as described in Section 3.2 and explained below), we can train strong toolken embeddings that outperform popular tool-learning methods.

**Dataset.** KAMEL [30] is a question-answering dataset built with the facts in Wikidata. In line with ToolFormer [56], which uses its earlier version [51] as a benchmark to evaluate the tool use, we adopt KAMEL to evaluate the use of KB query tools. KAMEL contains knowledge about 243 relations from Wikidata, each of which is associated with a question template (e.g. winner_of: "*Who is the winner of [S]?*") to turn a fact in Wikidata into a question. We have 234 tools in total for this dataset. In order to analyze the performance provided with different numbers of tools, we create four subsets by sampling from the original test set. Each subset consists of questions related to different numbers of relations, corresponding to 30, 60, 100, and 234, respectively. The size of each subset is 500.

**Comparison methods.** We set up two different variants of our framework. (1) *ToolkenGPT (sup)*: We sample 200 examples per relation from the training set of KAMEL and train the toolken embeddings via supervised learning. This setting represents real-world scenarios where sufficient in-domain training data is available. (2) *ToolkenGPT (syn)*: In a more challenging setting where we assume in-domain training data is not available, we use the text description of each relation to synthesize training data with ChatGPT, e.g. "*The Nobel Peace Prize in 2020 was awarded to the United Nations World Food Programme for its efforts...*", where the underlying tool call is winner_of(NOBEL PEACE PRIZE IN 2020)→UNITED NATIONS WORLD FOOD PROGRAMME. On average, 40 examples are used to train each toolken embedding.

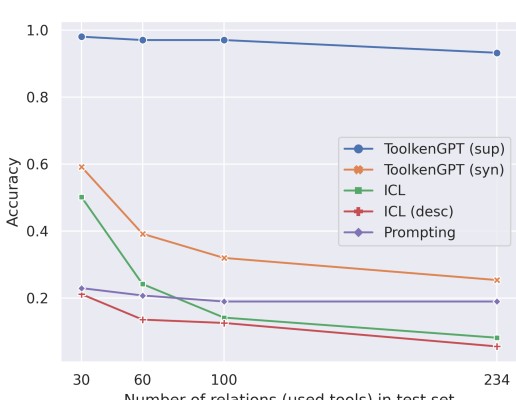

Figure 2: Performance of ToolkenGPT and baselines on 4 testsets involving different numbers of tools (relations) from KAMEL. ICL is short for In-context Learning [53]. Due to the context length limit of 2048 tokens, we list the descriptions and demonstrations of up to 30 relations for ICL and up to 60 relation descriptions for ICL (desc).

We introduce the following baselines for comparisons: (1) *Prompting* [30] is a straightforward method that answers the questions with the LLM's internal knowledge. We frame each question within the prompt "*Question: [QUESTION]\nThe answer is*" and ask the LLM to continue the sentence. (2) *In-context Learning (ICL)* [53] is a standard method to augment LLMs with tools as introduced in Section 2. Before asking the

question, we list the tool demonstrations and descriptions of all available tools. The demonstrations are shown in a specific syntax so that the LLM can generate in a similar style to be parsed. An example demonstration for `winner_of` is *"Question: Who is the winner of 2005-06 FA Cup?\nAnswer: The answer is <winner_of>(2005-06 FA Cup)=Liverpool F.C."* In a recent survey [53], this setting is referred to as "few-shot". (3) *In-context Learning (desc)* [53] is another common practice to augment LLMs with tools. The descriptions of all available tools will be provided in context, but their demonstrations are not directly shown. Instead, we show 8 demonstrations of the tools not included in the test subset to inform LLMs about the tool call format. This setting is referred to as "zero-shot" in Qin et al. [53]. The base model for all methods is LLaMA-13B [62]. More experiment details are described in Appendix B.

**Result analysis.** We show the experiment results on 4 testsets involving different numbers of relations in Figure 2. Note that the number of involved relations is the number of tools we can use. For all testsets, the accuracy of Prompting is about 20%, which indicates LLMs still struggle to store accurate facts in their parameters and it's necessary to augment them with a knowledge base. ToolkenGPT (sup) achieves the highest results with a large margin, showing that learning toolken embeddings is an effective method when there is massive in-domain training data. On the contrary, even though In-context learning also sees in-domain training data in the context, it still gets confused about which tools to call. Furthermore, the context length limit leads to drastic performance drops when there are more than 30 tools to use. The failure in the many-tools scene reveals the fundamental limitation of the in-context learning paradigm. ToolkenGPT (syn) also outperforms all other baselines in all subsets, without seeing any in-domain training data. The synthetic training data, often in very different expression styles from the dataset, still helps the LLM understand these relations.

This success reflects the flexibility of our framework which can be applied even if there is no in-domain training data available. In-context learning (desc) generally fails in this task, because the LLM has difficulties memorizing text descriptions shown in contexts and mapping them to relation identifiers. The results provide more evidence to the previous discovery that LLMs have trouble using unfamiliar tools [6]. Based on this observation, it is reasonable to speculate that LLMs mostly *recall* the tools from their identifier instead of really *learning* to use tools from their descriptions.

## 4.3 Embodied Plan Generation

Recently, there have been many research attempts to utilize LLMs as the controller of embodied agents [25, 59, 4, 27, 66]. Despite the preliminary success of prompting LLMs, teaching LLMs about an environment and enabling them to make grounded predictions remain challenging. As discussed in Mialon et al. [45], tools that gather additional information (e.g. math or KB tools) and tools that have an effect on the physical world (e.g. actions taken by embodied agents) can be called in similar styles by the LLM. In this section, we demonstrate how our framework can also be applied to plan generation for embodied agents. Compared to previous methods that prompt LLMs, our ToolkenGPT can understand the environment better by learning toolken embeddings for agent action and object.

**Dataset.** VirtualHome [52] is a simulation platform for typical household activities, and ActivityPrograms knowledge base [52] consists of many tasks with plans executable in VirtualHome. We derive a subset of 297 tasks from ActivityPrograms.

Specifically, for each task, the model is given a high-level **goal** (e.g. *"Read book"*), a detailed **instruction** (e.g. *"I would go lie down in my bed and open the book and start reading."*, and a description of the **environment**, which includes the initial state of the agent, and the object list of the environment (e.g. *"I am in ['home_office']. The objects I can manipulate are ['mail', 'freezer', 'television', ..., 'novel']"*. The model is expected to output an executable plan, which is an ordered list of verb-object instructions (e.g. *"[FIND] <novel>"*). Each task comes with an initial and final state graph, enabling the verification of the generated plans with the simulator and the comparison of the resulting final state with ground truth. We split the dataset into a training set of 247 tasks and a test set of 50 tasks, with a total of 25 verbs and 32 objects used in the dataset.

**Comparison methods.** We consider all the actions and objects in VirtualHome as tools. With an additional `[END]` function indicating the end of a plan, we have 58 toolkens in total. For this dataset, we do not need the argument generation process described in Figure 1 because the tools do not take arguments. During inference, ToolkenGPT alternatively generates action toolkens and object toolkens, and ends with the `[END]` toolken. The toolken embeddings are trained on the training set.

Table 3: Results on VirtualHome. Grounding means the proportion of scripts in which all the actions and objects can be grounded to the environment. Executable means the proportion of scripts that can be executed in VirtualHome without violating any rules. Success means the proportion of scripts that leads to the correct final state. Success (R) is a relaxed variant meaning the proportion of scripts that have reached the correct final state, but not necessarily ending with it.

| Method | Grounding | Executable | Success | Success (R) |
|---|---|---|---|---|
| In-context Learning | 0.74 | 0.42 | 0.20 | 0.30 |
| + Translation [25] | **1.00** | 0.52 | 0.24 | 0.32 |
| + Grounded Decoding [27] | **1.00** | 0.66 | 0.38 | 0.42 |
| ToolkenGPT (Ours) | **1.00** | **0.82** | **0.68** | **0.70** |

We compare our method to the following baselines: (1) *In-context Learning* prompts the LLM and parses its outputs as the plan. The LLM is shown with the action list, 3 demonstration plans, and a new task with its goal, detailed description, and environment description. This method is the base of most recent methods [25, 4, 27] that apply LLMs to embodied AI. (2) *Translation* [25]: To avoid plans that include unavailable actions or objects, Huang et al. [25] proposes to use a translation model to translate the LLM's generation to admissible instructions. Following Huang et al. [25], we use SentenceRoBERTa-large [54] and translate the actions or objects to available ones with the highest cosine similarities. (3) *Grounded Decoding* [27] is a recent decoding-stage grounding method. The next token is predicted considering both LLM logits and "grounded functions". Specifically, we apply the affordance grounding function [27], encouraging LLMs to generate valid actions and objects. We do not consider other previous methods that heavily fine-tune the whole language model [37]. The base model of all methods is LLaMA-13B [62]. More experiment details are described in Appendix C.

**Result analysis.** We list results in Table 3. Though all valid actions and objects are explicitly listed in the context for the LLM using In-context Learning, it sometimes fails to ground its prediction to admissible instructions. Even though the actions and objects are valid, they often violate the physical rule in VirtualHome, resulting in a low success rate. We notice that while most of the plans generated with In-context Learning appear reasonable to humans, they are not grounded to the specific environment of VirtualHome. Translation [25] helps solve some shallow grounding problems, e.g. [tv] → [television], while Grounded Decoding [27] further improves executable and success rate by considering grounding earlier in the decoding stage. Although these methods ensure

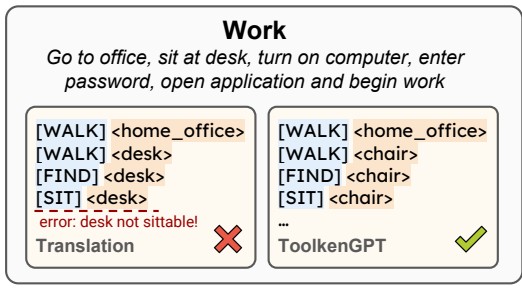

Figure 3: Case study on VirtualHome. ToolkenGPT predicts a successful script while other baselines fail to produce an executable one due to their misunderstanding of the SIT action.

all plans are grounded, neither significantly improves the LLM's understanding of actions and objects, leading to unsatisfactory executable and success rates. ToolkenGPT not only predict valid actions and objects naturally by its design, but also achieves the highest success rate by learning toolken embeddings from more training tasks. A concrete example is shown in Figure 3 to illustrate the difference: All the baselines predict [SIT] <desk>, presumably guided by the description "sit at desk", but in VirtualHome [SIT] refers to "sit on", and a desk is regarded as not sittable. ToolkenGPT is the only one to successfully learn this rule from demonstrations and instead predict [SIT] <chair>.

## 4.4 Analysis

**Computational Cost.** We conduct experiments to compare ToolkenGPT with fine-tuning, specifically using LoRA [23], in terms of computation efficiency and performance. Due to the cost of fine-tuning LLMs, we implement both methods on LLaMA-7B. The results are listed in Table 4.

Fine-tuning LLMs results in slightly better performance than ToolkenGPT on FuncQA. Even though we apply LoRA, which is known for efficiency, the time consumption for fine-tuning exceeds

Table 4: Comparison between ToolkenGPT and fine-tuning (LoRA) in terms of training cost and performance on FuncQA dataset. Both methods are based on Llama-7B.

| Method | One-hop | Multi-hop | Computing Resource | Training Time |
|---|---|---|---|---|
| ReAct | 0.40 | 0.03 | - | - |
| Prompting | 0.10 | 0.00 | - | - |
| Fine-tune w/ LoRA [23] | 0.62 | 0.07 | $8 \times$ A100 (80G) | 40 min |
| ToolkenGPT | 0.55 | 0.06 | $1 \times$ RTX3090 (24G) | 2 min |

Table 5: Ablation study on FuncQA dataset with LLaMA-30B.

| Method | One-hop | Multi-hop |
|---|---|---|
| ReAct | 0.57 | 0.06 |
| + Tool mode | 0.60 | 0.07 |
| ToolkenGPT | 0.73 | 0.15 |

Table 6: ToolkenGPT with different configurations of training data on KAMEL.

| # Examples | Synthetic | Supervised |
|---|---|---|
| 10 | 0.36 | 0.56 |
| 20 | 0.46 | 0.90 |
| 40 | 0.52 | 0.95 |

significantly when compared to training toolken embeddings. It is also worth noting that ToolkenGPT enjoys additional benefits other than efficiency (Table 1), especially the plug-and-play of massive tools, thanks to the decoupled parameters for different tools.

**Ablation Study.** The design of ToolkenGPT benefits both tool selection and argument completion (tool mode in Figure 1). To understand their respective contributions to the performance, we further implement a baseline combining ReAct-style prompting and the sub-routine of argument completion (tool mode). In the tool mode, the LLM is prompted with demonstrations using only the selected tool, which will provide more relevant knowledge than ReAct prompt for argument completion. As shown in Table 5, adding a tool mode could indeed improve the vanilla ReAct prompting method by enhancing the accuracy of argument completion. However, ToolkenGPT still outperforms this improved baseline by a large margin, indicating that toolken embeddings effectively help LLMs to decide when and which tool to call.

**Training Data.** In this section, we explore the effects of training data on learning the toolken embeddings. We choose to extend our experiments on KAMEL (Section 4.2), because there are two different sources of training data and it is easy to process or synthesize more data. Specifically, we sample 10/20/40 training examples of each tool for both ToolkenGPT (sup) and ToolkenGPT (syn), and report the accuracy on the test set involving 30 tools.

The results are summarized in Table 6. Under the same budget of data size, training with supervised data leads to better performance. Though we do not observe obvious mistakes in most of synthetic data instances, the distribution gap between synthetic data and test set may prevent toolken embedding from performing well. A larger training set benefits the performance for both data sources.

# 5 Conclusion

We presented ToolkenGPT, a new approach for augmenting frozen LLMs with massive external tools without expensive fine-tuning. Our method introduces the concept of toolken embedding for each tool, enabling LLMs to call and use different tools as easily as generating word tokens. Our approach overcomes the limitations of current fine-tuning and in-context learning paradigms, enabling LLMs to accommodate a much larger set of tools and use extensive demonstration data for learning toolken embeddings. On a series of tasks, ranging from numerical reasoning, knowledge-based question answering, to embodied plan generation, we observed a significant enhancement in LLM performance with the help of toolken embeddings. More importantly, ToolkenGPT is able to rapidly adapt and leverage new tools, demonstrating its capacity to keep pace with the constantly evolving landscape of massive tools. We expect future research to learn robust toolken embeddings not only from demonstration data, but also other rich forms of experience [24], such as tool descriptions and input-output records. We are also interested in exploring the integration of toolken embeddings to recent advanced planning techniques [19], with the goal of developing an autonomous agent to solve complex real-world problems.

## Acknowledgements

This project is partially supported by DARPA ECOLE HR00112390063.

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

# A  Details of Numerical Reasoning

In this section, we describe the data processing workflow (including building the testing datasets and synthesizing training data), list the prompts used for different methods, and describe the training setting. For a fair comparison, we use the same prompts for CoT and the reasoning mode of ToolkenGPT. With the same example questions, we label the calculation process in place to get the prompts for ReAct[3]. For the tool mode of ToolkenGPT, we randomly sample 4 examples of the specified tool from the training set, and transform them into ReAct-style prompts. Because a large number of tools are used, we show the prompts in the supplementary file instead of listing them here.

## A.1  GSM8K-XL

### A.1.1  Data Synthesis

To build an enhanced version GSM8K-XL by magnifying numbers in GSM8K, we take the following steps:

1. We prompt ChatGPT(`gpt-3.5-turbo`) with two examples to replace the numbers with appropriate placeholders. The prompt is presented below.

2. In order to validate the correctness of the number replacements, we develop a verification function for the GSM8k dataset, which calculates the formulas embedded in the solutions (See the example prompt below. Formulas are quoted with « and ».) to get an answer. By substituting the original numbers back into the rewritten question-answer pairs and comparing the execution results with the original answer, we can identify any erroneous rewrites and exclude them from our dataset. This ensures the reliability and accuracy of the number replacements.

3. Once we have confirmed the accuracy of the rewriting process, we can proceed with magnifying the numbers. We replace the original numbers with their respective cubic values, exponentially amplifying their magnitudes. However, it's worth noting this strategy doesn't make sense in every scenario. For instance, when dealing with percentage values (e.g., 10%), a direct cubic magnification to 1000% would be inappropriate. In such cases, we maintain the original values to preserve their intended meaning.

4. With the magnified numbers in place, we seamlessly substitute the placeholders within the rewritten question-answer pairs. By employing the solving function developed earlier, we can process the magnified formulas and obtain accurate solutions, resulting in the GSM8K-XL dataset.

```
{
    "role": "system",
    "content": "Your task is to rewrite the given question and answer in a
        more general form using variables instead of specific numbers."
},
{
    "role": "user",
    "content": "Question: Janet's ducks lay 16 eggs per day. She eats three
        for breakfast every morning and bakes muffins for her friends every
        day with four. She sells the remainder at the farmers' market daily
        for $2 per fresh duck egg. How much in dollars does she make every
        day at the farmers' market\nAnswer: Janet sells 16 - 3 - 4 =
        <<16-3-4=9>>9 duck eggs a day.\nShe makes 9 * 2 = $<<9*2=18>>18
        every day at the farmer's market.\n#### 18"
},
{
    "role": "assistant",
```

---

[3]As Yao et al. [69] doesn't evaluate ReAct on numerical reasoning, we don't follow the exact format (i.e. trajectories marked by `[THINK]` and `[ACT]`), but design a more natural way to combine CoT reasoning and tool calling (as shown in later sections). We refer to this format as "ReAct-style" prompts.

```
        "content": "Question: Janet's ducks lay {v_1} eggs per day. She eats
            {v_2} for breakfast every morning and bakes muffins for her friends
            every day with {v_3}. She sells the remainder at the farmers'
            market daily for ${v_4} per fresh duck egg. How much in dollars
            does she make every day at the farmers' market?\nAnswer: Janet
            sells {v_1} - {v_2} - {v_3} = <<{v_1}-{v_2}-{v_3}={r_1}>>{r_1} duck
            eggs a day.\nShe makes {r_1} * {v_4} = $<<{r_1}*{v_4}={r_2}>>{r_2}
            every day at the farmer's market.\n#### {r_2}"
},
{
    "role": "user",
    "content": "Question: A wooden bridge can carry no more than 5000
        pounds. A delivery truck filled with identical boxes, each weighing
        15 pounds, will pass over the bridge. The combined weight of the
        driver and the empty truck is 3755 pounds. What is the maximum
        number of boxes which can be loaded onto the truck while not
        exceeding the bridge's weight limit?\nAnswer: The boxes can weigh
        up to 5000 pounds - 3755 pounds = <<5000-3755=1245>>1245 pounds in
        total.\nThere can be 1245 / 15 = <<1245/15=83>>83 boxes loaded onto
        the truck without exceeding the bridge's weight limit.\n#### 83"
},
{
    "role": "assistant",
    "content": "Question: A wooden bridge can carry no more than {v_1}
        pounds. A delivery truck filled with identical boxes, each weighing
        {v_2} pounds, will pass over the bridge. The combined weight of the
        driver and the empty truck is {v_3} pounds. What is the maximum
        number of boxes which can be loaded onto the truck while not
        exceeding the bridge's weight limit?\nAnswer: The boxes can weigh
        up to {v_1} pounds - {v_3} pounds = <<{v_1}-{v_3}={r_1}>>{r_1}
        pounds in total.\nThere can be {r_1} / {v_2} =
        <<{r_1}/{v_2}={r_2}>>{r_2} boxes loaded onto the truck without
        exceeding the bridge's weight limit.\n#### {r_2}"
},
{
    "role": "user",
    "content": [INPUT]
}
```

### A.1.2  Training Details

As mentioned in Section 4.1, the toolken embeddings are trained with a subset of 5,063 examples. An additional 1,000 examples are reserved for validation. The embeddings were trained with a learning rate of 5e-4, performing early stopping based on the development set, with a maximum of 10 epochs.

### A.1.3  Prompt for GSM8K-XL Dataset

Prompt for Direct Prompting with ChatGPT:

```
Solve the following math problem step by step, and then provide the final
answer in the format: 'So, the answer is xxx.'

[QUESTION]
```

Prompt for Chain of Thought (CoT) and ToolkenGPT reasoning mode:

```
Answer the following questions step by step.

Question: Mark has 3 tanks for pregnant fish. Each tank has 4 pregnant
fish and each fish gives birth to 20 young. How many young fish does he
have at the end?
```

Answer: He has 4*3=12 pregnant fish They give birth to 12*20=240 fish ####
240

Question: The math questions in a contest are divided into three rounds:
easy, average, and hard. There are corresponding points given for each
round. That is 2, 3, and 5 points for every correct answer in the easy,
average, and hard rounds, respectively. Suppose Kim got 6 correct answers
in the easy; 2 correct answers in the average; and 4 correct answers in
the difficult round, what are her total points in the contest?
Answer: Kim got 6 points/round x 2 round = 12 points in the easy round.
She got 2 points/round x 3 rounds = 6 points in the average round. She got
4 points/round x 5 rounds = 20 points in the difficult round. So her total
points is 12 points + 6 points + 20 points = 38 points. #### 38

Question: A clothing store sells 20 shirts and 10 pairs of jeans. A shirt
costs $10 each and a pair of jeans costs twice as much. How much will the
clothing store earn if all shirts and jeans are sold?
Answer: Twenty shirts amount to $10 x 20 = $200. The cost of each pair of
jeans is $10 x 2 = $20. So 10 pairs of jeans amount to $20 x 10 = $200.
Therefore, the store will earn $200 + $200 = $400 if all shirts and jeans
are sold. #### 400

Question: Arnold's collagen powder has 18 grams of protein for every 2
scoops. His protein powder has 21 grams of protein per scoop. And his
steak has 56 grams of protein. If he has 1 scoop of collagen powder, 1
scoop of protein powder and his steak, how many grams of protein will he
consume?
Answer: 2 scoops of collagen powder have 18 grams of protein and he only
has 1 scoop so he consumes 18/2 = 9 grams of protein He has 9 grams
collagen powder, 21 grams of protein powder and 56 grams in his steak for
a total of 9+21+56 = 86 grams of protein #### 86

Question: [QUESTION]
Answer:

Prompt for ReAct:

Answer the following questions with <add>, <subtract>, <multiply>,
<divide> operators

Question: Mark has 3 tanks for pregnant fish. Each tank has 4 pregnant
fish and each fish gives birth to 20 young. How many young fish does he
have at the end?
Answer: He has 4*3=<multiply>(4, 3)=12 pregnant fish They give birth to
12*20=<multiply>(12, 20)=240 fish #### 240

Question: The math questions in a contest are divided into three rounds:
easy, average, and hard. There are corresponding points given for each
round. That is 2, 3, and 5 points for every correct answer in the easy,
average, and hard rounds, respectively. Suppose Kim got 6 correct answers
in the easy; 2 correct answers in the average; and 4 correct answers in
the difficult round, what are her total points in the contest?
Answer: Kim got 6 points/round x 2 round = <multiply>(6, 2)=12 points in
the easy round. She got 2 points/round x 3 rounds = <multiply>(2, 3)=6
points in the average round. She got 4 points/round x 5 rounds =
<multiply>(4, 5)=20 points in the difficult round. So her total points is
12 points + 6 points + 20 points = <add>(12, 6, 20)=38 points. #### 38

```
Question: A clothing store sells 20 shirts and 10 pairs of jeans. A shirt
costs $10 each and a pair of jeans costs twice as much. How much will the
clothing store earn if all shirts and jeans are sold?
Answer: Twenty shirts amount to $10 x 20 = $<multiply>(10, 20)=200. The
cost of each pair of jeans is $10 x 2 = $<multiply>(10, 2)=20. So 10 pairs
of jeans amount to $20 x 10 = $<multiply>(20, 10)=200. Therefore, the
store will earn $200 + $200 = $<add>(200, 200)=400 if all shirts and jeans
are sold. #### 400

Question: Arnold's collagen powder has 18 grams of protein for every 2
scoops. His protein powder has 21 grams of protein per scoop. And his
steak has 56 grams of protein. If he has 1 scoop of collagen powder, 1
scoop of protein powder and his steak, how many grams of protein will he
consume?
Answer: 2 scoops of collagen powder have 18 grams of protein and he only
has 1 scoop so he consumes 18/2 = <divide>(18, 2)=9 grams of protein He
has 9 grams collagen powder, 21 grams of protein powder and 56 grams in
his steak for a total of 9+21+56 = <add>(9, 21, 56)=86 grams of protein
#### 86

Question: [QUESTION]
Answer:
```

## A.2 FuncQA

### A.2.1 Training Details

As mentioned in Section 4.1, Toolken embeddings are trained using a subset of 611 examples, with
an additional 39 examples reserved for validation purposes. The learning rate we use is 1e-4, and we
perform early stopping based on the development set, with the maximal training epochs to be 20.

### A.2.2 Prompt for Synthetic Training Data

In Section 4.1, we discussed the utilization of ChatGPT for synthesizing the training set. To create
the training data, we begin by manually crafting two examples that adhere to the desired format,
and then use the following specific prompt to generate more examples. However, it is important to
acknowledge that the prompt does not guarantee the generation of examples that strictly conform
to the required format. So, we apply a filtering process to remove any non-conforming instances.
Furthermore, the generation process often produces duplicate examples, necessitating a subsequent
de-duplication step.

```
You are a math question generator for teachers, and your task is to
generate some questions and answers using function [FUNC] to solve and can
be solved within one single step. You do not need to give specific
numbers, so that the teachers can fill any numbers they want. Here are two
examples that use the function [FUNC].

[EXAMPLE_1]

[EXAMPLE_2]

[FUNC] is a function to [DESCRIPTION]. Now, let's mimic the format of
examples to generate various real world QA pairs using the function [FUNC]
that can be solved within one step. The numbers should be replaced by
[ARG] and [ANSWER] as the examples given above.
```

### A.2.3 Prompt for FuncQA One-Hop

Prompt for Zero-Shot with ChatGPT:

Solve the following math problem, and then provide the final answer in the format: 'So, the answer is xxx.'

[QUESTION]

Prompt for Chain of Thought (CoT) and ToolkenGPT:

Q: If Amy's income increases by 4% annually, how many times will it multiply in 11 years?
A: In 11 years, Amy's income will increase by 1.04^11=1.54 times. So, the answer is 1.54.

Q: If a store sells 147 bananas today and 354 more bananas tomorrow, how many bananas does the store sell in total?
A: The store sells 147 bananas today and 354 more bananas tomorrow, so the total number of bananas sold is 147+354=501. So, the answer is 501.

Q: A man had 789.4 dollars in his wallet. He spent 11.99 dollars on a movie ticket. How much money does he have left now?
A: The man had 789.4 dollars in his wallet and spent 11.99 dollars on a movie ticket, so he has 789.4-11.99=777.41 dollars left. So, the answer is 777.41 dollars.

Q: If a cake weighs 3.77 pounds and is divided into 13 equal pieces, how much does each piece weight?
A: Each piece of the cake weighs 3.77/13=0.29 pounds. So, the answer is 0.29 pounds.

Q: [QUESTION]
A:

Prompt for ReAct:

Answer the following question with <add>, <subtract>, <multiply>, <divide>, <power>, <sqrt>, <log>, <lcm>, <gcd>, <ln>, <choose>, <remainder>, <permutate>:

Q: If Amy's income increases by 4% annually, how many times will it multiply in 11 years?
A: In 11 years, Amy's income will increase by 1.04^11 = <power>(1.04,11)=1.54 times. So, the answer is 1.54.

Q: If a store sells 147 bananas today and 354 more bananas tomorrow, how many bananas does the store sell in total?
A: The store sells 147 bananas today and 354 more bananas tomorrow, so the total number of bananas sold is 147+354=<add>(147,354)=501. So, the answer is 501.

Q: A man had 789.4 dollars in his wallet. He spent 11.99 dollars on a movie ticket. How much money does he have left now?
A: The man had 789.4 dollars in his wallet and spent 11.99 dollars on a movie ticket, so he has 789.4-11.99=<subtract>(789.4,11.99)=777.41 dollars left. So, the answer is 777.41.

Q: If a cake weighs 3.77 pounds and is divided into 13 equal pieces, how much does each piece weight?
A: Each piece of the cake weighs 3.77/13=<divide>(3.77,13)=0.29 pounds. So, the answer is 0.29.

Q: [QUESTION]

```
A:
```

### A.2.4  Prompt for FuncQA Multi-Hop

Prompt for Zero-Shot with ChatGPT:

```
Solve the following math problem step by step, and then provide the final
answer in the format: 'So, the answer is xxx.'

[QUESTION]
```

Prompt for Chain of Thought (CoT) and ToolkenGPT:

```
Answer the following questions step by step:

Question: A coin is tossed 8 times, what is the probability of getting
exactly 7 heads ?
Answer: The total number of possible outcomes to toss a coin 8 times is
2^8=256. The number of ways of getting exactly 7 heads is 8C7=8. The
probability of getting exactly 7 heads is 8/256=0.03125. #### 0.03125

Question: If paint costs $3.2 per quart, and a quart covers 12 square
feet, how much will it cost to paint the outside of a cube 10 feet on each
edge?
Answer: The total surface area of the 10 ft cube is 6*10^2=6*100=600
square feet. The number of quarts needed is 600/12=50. The cost is
50*3.2=160. #### 160

Question: log(x)=2, log(y)=0.1, what is the value of log(x-y) ?
Answer: log(x)=2, so x=10^2=100; log(y)=0.1, so y=10^0.1=1.26;
x-y=100-1.26=98.74, so log(x-y)=log(98.74)=1.99. #### 1.99

Question: How many degrees does the hour hand travel when the clock goes
246 minutes?
Answer: The hour hand travels 360 degrees in 12 hours, so every hour it
travels 360/12=30 degrees. 246 minutes is 246/60=4.1 hours. The hour hand
travels 4.1*30=123 degrees. #### 123

Question: [QUESTION]
Answer:
```

Prompt for ReAct:

```
Answer the following questions with <add>, <subtract>, <multiply>,
<divide>, <power>, <sqrt>, <log>, <lcm>, <gcd>, <ln>, <choose>,
<remainder>, and <permutate>:

Question: A coin is tossed 8 times, what is the probability of getting
exactly 7 heads?
Answer: The total number of possible outcomes to toss a coin 8 times is
2^8=<power>(2,8)=256. The number of ways of getting exactly 7 heads is
8C7=<choose>(8,7)=8. The probability of getting exactly 7 heads is
8/256=<divide>(8,256)=0.03125. #### 0.03125

Question: If paint costs $3.2 per quart, and a quart covers 12 square
feet, how much will it cost to paint the outside of a cube 10 feet on each
edge?
Answer: The total surface area of the 10 ft cube is
6*10^2=6*<power>(10,2)=100=<multiply>(6,100)=600 square feet. The number
of quarts needed is 600/12=<divide>(600,12)=50. The cost is
50*3.2=<multiply>(50,3.2)=160. #### 160
```

```
Question: log(x)=2, log(y)=0.1, what is the value of log(x-y) ?
Answer: log(x)=2, so x=10^2=<power>(10,2)=100; log(y)=0.1, so
y=10^0.1=<power>(10,0.1)=1.26; x-y=100-1.26=<subtract>(10,1.26)=98.74, so
log(x-y)=log(98.74)=<log>(98.74)=1.99. #### 1.99

Question: How many degrees does the hour hand travel when the clock goes
246 minutes?
Answer: The hour hand travels 360 degrees in 12 hours, so every hour it
travels 360/12=<divide>(360,12)=30 degrees. 246 minutes is
246/60=<divide>(246,60)=4.1 hours. The hour hand travels
4.1*30=<multiply>(4.1,30)=123 degrees. #### 123

Question: [QUESTION]
Answer:
```

# B  Details of Knowledge-based QA

In this section, we show how to transform each Wikidata relation identifier (e.g. P1346) into a natural language description [4], and then describe the method to synthesize data and the training settings.

## B.1  Getting Text Description

KAMEL provides a question template for each relation. We randomly sample 3 facts from the dataset and instantiate them into question-answer pair, and use the following prompt to generate a description for them with ChatGPT:

```
Given a question template and some example answer, you need to define an
API that can help you answer the question.
Q 1.1: What is the original language of The Wonderful Galaxy of Oz
A 1.1: Japanese
Q 1.2: What is the original language of Wild Field?
A 1.2: Russian
Q 1.3: What is the original language of Nadigan?
A 1.3: Tamil
API 1: original_language(title): gets the original language of an art work
Q 2.1: What languages does Judah Maccabee speak?
A 2.1: Hebrew
Q 2.2: What languages does Ronelda Kamfer speak?
A 2.2: Afrikaans
Q 2.3: What languages does Leibush Lehrer speak?
A 2.3: Yiddish
API 2: spoken_languages(name): gets the spoken languages of a person
Q 3.1: [Q1]
A 3.1: [A1]
Q 3.2: [Q2]
A 3.2: [A2]
Q 3.3: [Q3]
A 3.3: [A3]
API 3:
```

## B.2  Synthetic Data

We use two prompts to synthesize diverse training data, and aggregate the samples from each prompt.

---

[4]Note that the natural language descriptions are not necessary for ToolkenGPT which is directly trained with demonstrations, but they are crucial for the in-context learning baselines, especially for ICL (desc), which can only understand tools through language descriptions.

```
Here are some examples of using functions for text generation (after the
function call, the sentence should continue with the returned value of the
function call):
1. star_rating(product): gets the star rating of the product on a scale
from 0 to 5.
Example 1.1: The new iPhone 12 Pro Max is already generating a lot of
buzz, thanks to its <f>star_rating("iPhone 12 Pro Max")="4.7"</f>4.7 star
rating.
2. literary_genre(book): gets the literary genre of a book
Example 2.1: Literature is often categorized by genre, such as drama,
romance, or science fiction. The Harry Potter series is a popular example
of the <f>literary_genre("Harry Potter")="fantasy"</f>fantasy genre, which
features imaginary worlds and magical elements.
3. current_location(user_id): gets the current location of a user.
Example 3.1: If you're trying to coordinate plans with a friend, it's
helpful to know their current location. You can ask the question "Where
are you right now?" and use the function <f>current_location("1234")="New
York"</f>New York as an example response.
4. number_of_movies(director): gets the number of movies directed by a
specific director.
Example 4.1: Martin Scorsese is one of the most celebrated movie directors
of all time. He has directed a total of <f>number_of_movies("Martin
Scorsese")="78"</f>78 movies throughout his career.
5. word_definition(word): gets the definition of a particular word
Example 5.1: Writers and English language learners can enhance their
vocabulary by knowing the definition of unfamiliar words. The definition
of the word "eccentric" is <f>word_definition("eccentric")="unconventional
and slightly strange"</f>unconventional and slightly strange.
6. number_of_spotify_followers(artist): gets the number of Spotify
followers for the artist.
Example 6.1: Taylor Swift's latest album is a hit and her fan base is
growing rapidly. In fact, her number of Spotify followers as of today is
<f>number_of_spotify_followers("Taylor Swift")="49,879,220"</f>49,879,220.
6. [DESCRIPTION]
Please continue to generate 10 examples using the function [NAME],
starting with 6.1 to 6.10.
```

```
Here are some examples of using functions for text generation (after the
function call, the sentence should continue with the returned value of the
function call):
1. current_weather(city): gets the current weather of a city.
Example 1.1: What's the weather in Beijing now? The weather is
<f>current_weather("Beijing")="sunny"</f>sunny now. Example 1.2: Do you
know what's the weather in San Diego now? The weather is
<f>current_weather("San Diego")="cloudy"</f>cloudy now.
2. calculator(formula): gets the calculation result of a formula.
Example 2.1: What's sum of 213 and 5032? The answer is
<f>calculator("213+5032")="5245"</f>5245.
Example 2.2: What's difference between 2015 and 33? The answer is
<f>calculator("2015-33")="1982"</f>1982.
3. [DESCRIPTION]
Please continue to generate 10 examples using the function [NAME],
starting with 3.1 to 3.10.
```

## B.3 Training Details

Toolken embeddings are trained with a learning rate of 1e-4, performing early stopping based on the development set, and trained for a maximum of 5 epochs.

# C  Details of Embodied Plan Generation

In this section, we describe the preprocessing of VirtualHome, and list all the prompts and training details.

## C.1  Preprocessing

We collect all scripts from ActivityPrograms [25] and filter the dataset with the following steps: (1) filter out all the scripts that are not executable, or don't cause any state changes in VirtualHome (2) deduplicate the scripts with the same goal and instruction. (3) discard the script that involves two different objects of the same name (4) find the verbs and objects that occur more than 10 times in the data, and keep the scripts composed of only these verbs and objects.

Note that our preprocessing is different from Huang et al. [25], where they regard a high-level goal as a task. We treat two scripts with the same goal but different instructions as distinct tasks because different instructions often indicate different action sequences, which may lead to different final state graphs, e.g., for a high-level goal "Reading", some of the instructions mention "Turn on desk lamp" while others don't. Huang et al. [25] relies on human annotation to evaluate the correctness of the generated script, which actually lowers the difficulties of learning the environment, because humans may assign a correct label as long as the plan looks "reasonable". On the contrary, we can use the Evolving Graph [5] to strictly match the resulting state and ground truth state. This serves as an automatic and more objective evaluation.

## C.2  Prompts

We show the prompts for LLMs to generate plans below. Note that all methods use the same prompts in this experiment.

```
I am a household robot and I can take actions from '[FIND]', '[SIT]',
'[SWITCHON]', '[TURNTO]', '[LOOKAT]', '[TYPE]', '[WALK]', '[LIE]',
'[GRAB]', '[READ]', '[WATCH]', '[POINTAT]', '[TOUCH]', '[SWITCHOFF]',
'[OPEN]', '[PUSH]', '[PUTOBJBACK]', '[CLOSE]', '[DRINK]', '[RUN]',
'[DROP]', '[PULL]'.

Task 1:
I am in ['bathroom']. The objects I can manipulate are ['faucet',
'keyboard', 'television', 'coffe_maker', 'chair', 'button', 'pillow',
'phone', 'cup', 'couch', 'freezer', 'desk', 'oven', 'light', 'table',
'bedroom', 'dining_room', 'cupboard', 'computer', 'sink', 'mail', 'bed',
'mouse', 'home_office'].
Goal:
Write an email
Hint:
i went near the computer and turned it on. then sent the mail
Plan:
[WALK] <home_office>
[WALK] <table>
[FIND] <table>
[WALK] <table>
[FIND] <computer>
[TURNTO] <computer>
[LOOKAT] <computer>
[TURNTO] <computer>
[SWITCHON] <computer>
[FIND] <mail>
[TURNTO] <mail>

Task 2:
```

---

[5] https://github.com/xavierpuigf/virtualhome

```
I am in ['home_office']. The objects I can manipulate are ['faucet',
'novel', 'keyboard', 'television', 'newspaper', 'chair', 'coffe_maker',
'pillow', 'phone', 'check', 'couch', 'freezer', 'desk', 'toothbrush',
'oven', 'light', 'food_food', 'table', 'bookmark', 'bedroom',
'dining_room', 'computer', 'sink', 'mail', 'bed', 'cat', 'mouse',
'home_office', 'pot'].
Goal:
Work
Hint:
Find the computer. Turn it on by pressing the on button. Wait for it to
load. Use the mouse and keyboard to perform your tasks on screen.
Plan:
[FIND] <computer>
[SWITCHON] <computer>
[FIND] <mouse>
[TOUCH] <mouse>
[FIND] <keyboard>
[TOUCH] <keyboard>

Task 3:
I am in ['bathroom']. The objects I can manipulate are ['dishwasher',
'faucet', 'keyboard', 'television', 'newspaper', 'chair', 'coffe_maker',
'pillow', 'phone', 'cup', 'check', 'couch', 'freezer', 'desk', 'oven',
'light', 'food_food', 'plate', 'table', 'bookmark', 'bedroom',
'dining_room', 'cupboard', 'computer', 'sink', 'bed', 'cat', 'mouse',
'home_office', 'pot'].
Goal:
Pick up phone
Hint:
first when i hear the ringing sound i will run to my living room and picks
up and i will say hello
Plan:
[RUN] <home_office>
[WALK] <chair>
[FIND] <chair>
[SIT] <chair>
[FIND] <phone>
[GRAB] <phone>

Task 4:
[QUESTION]
```

## C.3 Training Details

Toolken embeddings are trained with a learning rate of 1e-4, performing early stopping based on the development set, with a maximum of 10 epochs.

## D   Computational Resources

In terms of computational resources, we train and test ToolkenGPT based on LLaMA-13B and LLaMA-33B using 2 and 4 Nvidia RTX 3090 GPUs, respectively.

## E   Safeguard Statement

In this paper, we primarily focus on the applications of mathematical, knowledge-based, and embodied planning problems, posing no significant ethical or harmful concerns. We recognize that future research on border applications of tool learning may pose a risk of misuse, and we recommend careful consideration of all aspects of safety before it's applied in the real world.

