# OpenReview forum: "ToolkenGPT: Augmenting Frozen Language Models with Massive Tools via Tool Embeddings"
_NeurIPS.cc/2023/Conference — NeurIPS 2023 oral_

### Official Review · Reviewer_QLCo · 2023-06-10

**Soundness:** 3 good
**Presentation:** 3 good
**Contribution:** 3 good
**Rating:** 7
**Confidence:** 4

**Summary:**

The authors propose "ToolkenGPT." This method allows language models to take advantage of external tools without full model fine-tuning, and with the ability to show more tools and with greater detail than in-context learning typically allows. They show that use of "toolkens" (tool tokens) improves performance over baselines on arithmetic, knowledge-base QA, and embodied plan generation.

**Strengths:**

In terms of originality, the paper introduces a new and innovative approach to external tool access for language models. It seems like a hybrid of fine-tuning and few-shot learning, but uses both in a novel and interesting way. Though there's not much direct comparison to prior work, it is at least acknowledged.

In terms of quality, the experimental setup is sound. It seems evident that ToolkenGPT performs better than simple (but well designed) baselines.

In terms of clarity, the paper is well-written and easy to follow. I haven't read the appendix in detail, but it seems to provide enough information for reproducing the experiments.

In terms of significance, this work brings up new ideas that seem very useful.


**Weaknesses:**

* The biggest weakness of this paper is in evaluation.
    * ToolkenGPT is not directly compared to prior work. It is only compared to things along the lines of zero-shot or few-shot baselines. At the end of the paper I'm left wondering how ToolkenGPT would stack up against e.g., a full model fine-tuning method like Toolformer. I can see that ToolkenGPT allows for demonstrating more tools with more detail than few-shot. It also seems like it's probably cheaper to use ToolkenGPT than to fine-tune. But I'm not really sure of how accuracy for the two tuning methods compare.
    * In evaluation there are sometimes confounding variables that render results less useful. For example, in Section 4.2 5x more data is used for training ToolkenGPT (sup) than ToolkenGPT (syn). Is the improved performance due to more training samples or higher data quality?
    * Evaluation sometimes lacks specificity. For example, in Section 4.2 it's unclear whether the 30 relations ICL has access to in the e.g., 234 relation case are required to include the correct relation. That seems like a very important detail.
    * Overall this paper covers an impressive breadth of evaluation, but I wish there was one in-depth evaluation where # of tools, model size, and number of training data points were carefully considered for strong few-shot and fine-tuning (e.g., fine-tuning on all the exact data used to train the toolkens) baselines along with ToolkenGPT while carefully controlling for other variables.
* I think the comparison between fine-tuning, few-shot, and ToolkenGPT could cleaner
    * I'm not sure of the difference between "plug-and-play" and "frozen LM" in Table 1. I wouldn't say ToolkenGPT is "plug-and-play" given the model-specific training required for embeddings.
    * I'm not sure what is really meant by "massive tools." What is stopping a fine-tuning method from learning such tools?

**Questions:**

See weaknesses

**Limitations:**

The authors have a nice discussion of limitations.

---

> ### Author Rebuttal · Authors · 2023-08-10
>
> We thank reviwer QLCo for acknowledging that our approach is new, innovative and interesting, brining up new and useful ideas to the field. Below we address all the reviewer's concerns and questions.
>
> (a) Comparing with full-model fine-tuning. We appreciate that the reviewer acknowledge that our proposed ToolkenGPT allows for more tools with more details than few-shot and is much cheaper than fine-tuning. We agree that a comaprison with full-model fine-tuning would bolster our evaluation and help clarify the unique advantages of our method. We explicitly address such an important comparison in the global response (b).
>
> (b) **Training data for ToolkenGPT (sup) and ToolkenGPT (syn)**. We appreciate the great observation from the reviewer and provide further experiment results to investigate the influence on training sample quantity and quality. Specifically, we sample 10/20/40 training examples of each tool for both ToolkenGPT (sup) and ToolkenGPT (syn), and report the accuracy on the 30-tools testset.
>
> RTable 5: Influence of training quantity and quality on ToolkenGPT.
>
> | # Examples per tool | ToolkenGPT (syn) | ToolkenGPT (sup) |
> | -------- | -------- | -------- |
> | 10     | 0.364     | 0.564     |
> | 20     | 0.464     | 0.896     |
> | 40     | 0.524     | 0.948     |
>
> RTable 5 shows the performance of ToolkenGPT (sup) and ToolkenGPT (syn) under the same amount of training data. As we can see, both the size and source of training data matter here, and the supervised data clearly contribute to the higher performance in the Figure 2 of the paper. We will include such ablation study into the revised version and we hope this could clarify the question from the reviewer.
>
> (c\) **Tool selection for ICL baseline**. We appreciate the reviewer for mentioning this very important detail. To illustrate this process clearly, let's index all the relations from 1 to 234. As explained in Line 273 of the paper, we have 4 datasets that involve the relations 1-30 (30-relation dataset), 1-60 (60-relation dataset), 1-100 (100-relation dataset), 1-234 (234-relation dataset), respectively.
>
> - The prompt for `ICL` includes the descriptions and the demonstrations of relations 1-30.
> - The prompt for `ICL (desc)` inclues the descriptions of relations 1-30 (for the first dataset) or 1-60 (for other datasets), and demonstrations of 8 random relations.
>
> It's worth noting that, even in the cases where `ICL` / `ICL (desc)` has access to every helpful tools for the dataset, i.e., for the 30-relations dataset, the performance is still inferior to ToolkenGPT (see Figure 2, datapoints in the first column). This indicates that even if the context length limit is resolved, in-context learning methods could still struggle to choose from massive tools. We hope this will clarify the question from the reviewer and we will clarify this detail clearly in the revised version.
>
> (d) **Comparison between fine-tuning, few-shot and ToolkenGPT**. We thank the reviewer for brining up this clarification question. First, we refer the reviewer to the global response (a) for a comprehensive comparison with fine-tuning. We also address the comment on the "plug-and-play" capaciy of ToolkenGPT in the global response (b). To sum up, we understand the reviewer's concern on the overlapping between "frozen LMs" and "plug-and-play". The aspect of "frozen LMs" in our paper focuses on the training efficiency, while "plug-and-play" is more challenging and requires more elegant design of parameters/prompts to reuse previous tools and plug them into new tasks.
>
> Regarding the definition of "Massive tools", we mean a large collection of tools that our model can learn and utilize. We agree that fine-tuning methods can learn such tools, but the innovation in our work is the ability to learn them with quick adaption and without exhaustive fine-tuning. When massive tools are presented, fine-tuning requires more training data (because we have more parameters to optimize) and more computing resources, which is costly to obtain, while our ToolkenGPT can "scales well to a massive number of tools with limited examples" mentioned by R-C7TV. More importantly, an implicit characteristic of "massive tools" is the dynamics of tools such that new tools are kept invented and updated every day. Thus, the growing massive tools make fine-tuning prohibitive to update model parameters frequently. We hope such a more detailed explanation could resolve the reviewer's concern and we will definitely clarify such a comparison more clearly in the revised version.

---

> > ### Comment · Reviewer_QLCo · 2023-08-15
> >
> > With respect to (a) I think including this line of experiment into the paper will be important. It seems like based on RTable 1 the conclusion is  that full fine-tuning reaches better accuracy, but at a high computation cost. It would be nice for the final version of the paper to emphasize the computation cost vs accuracy trade-off.
> >
> > Thanks for clarifying (b) and (c). I think including that information in the paper will be useful.
> >
> > I'm happy you're planning to revise/clarify the "plug and play" term (d). While it's perhaps a small point, I still find the term "Massive tools" a bit misleading. "Massive tools" seems to imply the tools are individually complex. But it seems like what you're trying to emphasize is that the model can handle many tools (regardless of their individual complexity). Anyways, it's a small thing but maybe worth some thought.
> >
> > Overall, thanks for the additional experiments. I trust the new results/clarifications will be added to the final version of the paper and I have accordingly recommended acceptance of the paper.

---

### Official Review · Reviewer_5YmU · 2023-07-03

**Soundness:** 4 excellent
**Presentation:** 3 good
**Contribution:** 4 excellent
**Rating:** 8
**Confidence:** 4

**Summary:**

This paper proposes a new method to help LLM utilize tools. Essentially, the method trains an additional output embedding token (toolken) for each tool. During inference, if LLM outputs a toolken, a special routine of using the corresponding tool is triggered and the returned result of the tool is then replaced with the toolken for continual inference. As a result, the proposed method can accommodate more tools and unfamiliar tools, compared to in-context prompting approaches like ReAct. The author has shown the effectiveness and advantage of the proposed method across multiple domains including math reasoning, knowledge based QA and embodied AI.

**Strengths:**

1. Compared to full model finetuning like Toolformer or TALM, the proposed method is much more efficient as it doesn't require gradient of LLM parameters and only tunes an additional output embedding matrix.
2. Empirical results on multiple domains verified the advantage of ToolkenGPT over ReAct, as it can better accommodate more tools.
3. ToolkenGPT is easy to implement and is able to be applied for various domains, which can potentially foster more research on the direction of enhancing LLMs with tools.

**Weaknesses:**

1. several typos are present: a) line 2, missing period after "problems"; b) line 92, missing space "whichhelps"; c) line 204, versatile *and* adaptive; d) line 234, 503.2 --> 50*3.2
2. One limitation is ToolkenGPT requires access to model activations thus not capable of being applied on closed-sourced LLMs like ChatGPT, while pure prompting method like ReAct doesn't have such limitation. As such, it's desirable to compare with a baseline of ReAct w/ ChatGPT. However, in the experiments, the author only compare with ChatGPT with 0-shot prompting (Table 2)
3. There are two parts of the proposed methods: toolken embedding and sub-routine of tool call. It's desirable to show some ablation of the latter part. For example, show a baseline of ReAct + sub-routine of tool call with demonstrations. In this way, readers can better assess the benefit of each component of proposed method

**Questions:**

1. The paper did not provide empirical comparison between ToolkenGPT with finetuning approach like Toolformer and TALM. Though you mentioned that ToolkenGPT requires much less GPU memory than those methods, is it possible to provide some empirical comparison to those methods?

**Limitations:**

The author has adequately discussed the limitations.

---

> ### Author Rebuttal · Authors · 2023-08-10
>
> We thank Reviewer 5YmU for acknowledging that our proposed method is much more efficient, easy to implement, and can be applied to various domains to foster more research in the field of tool-augmented LLMs. We also thank the reviewer for spotting several typos and we will fix them all in the revised version. Below we resolve the reviewer's concerns and questions.
>
> (a) **Baseline of ReAct w/ ChatGPT**. We understand the reviewer's concern regarding the potential limitation of ToolkenGPT in its inability to be applied on closed-sourced LLMs like ChatGPT. From the perspective of the open-sourced community, we agree that a comparison with a baseline of ReAct w/ ChatGPT would provide a broader context and help further validate the efficacy of our method. As shown in Table 2, we already compare our method with ReAct but based on the same based model (LLaMA-30B), mainly for a fair comparison. As ChatGPT is a much stronger base model than LLaMA-30B, applying ReAct to ChatGPT is expected to boost the performance. However, our ToolkenGPT should also be benefited from stronger open-sourced LLMs, like the recent Llama 2 models, which we believe should be an interesting future work to explore.
>
>
> (b) **Ablation study**. Thanks for the great suggestion! To better assess the benefit of each component of ToolkenGPT, we conduct an ablation study for the two main components of our method - toolken embedding and sub-routine of tool call (tool mode) as follows.
>
> RTable 4: Ablation study on tool embeddings and tool mode.
>
> | Method | FuncQA (one-hop) | FuncQA (multi-hop) |
> | -------- | -------- | -------- |
> | ReAct     | 0.57     | 0.06     |
> | ReAct + tool mode     | 0.60     | 0.07     |
> | ToolkenGPT | 0.73 | 0.15 |
>
> RTable 4 is closely related to Table 2 from the paper, where we use a hard math dataset, FuncQA with 13 math tools. We further implement a baseline of combing ReAct prompt and the sub-routine of tool call (tool mode) from our ToolkenGPT. As shown in this table, the tool mode used in ToolkenGPT could help improve the correctness of tool calling and marginally improve the vanilla ReAct prompting method. We also see that ToolkenGPT still outperforms this improved baseline by a large margin, indicating that our toolken embeddings still outperform ReAct greatly in terms of deciding when and which tool to call. We hope these additional results could help provide clear evidence of the contribution of each component to the performance of ToolkenGPT.
>
>
> (c\) Comparing with fine-tuning approaches. Please refer to the global response (b) for addressing the comparison with full-model fine-tuning methods, like Toolformer.

---

> > ### Comment · Reviewer_5YmU · 2023-08-15
> > **Response to Author's Rebuttal**
> >
> > Thank you for providing the detailed responses. I have read it carefully and hope the additional experiments will go into your future version.

---

### Official Review · Reviewer_Mpoh · 2023-07-04

**Soundness:** 4 excellent
**Presentation:** 4 excellent
**Contribution:** 3 good
**Rating:** 7
**Confidence:** 4

**Summary:**

This paper presents ToolkenGPT, a novel approach to generalizing LLMs to massive external tools. The key is to represent each tool as a "token", such that when LLMs generate the tool token(toolken), the associated tool will be invoked and executed. The learning of toolken is easy and efficient since there is no need to update the whole LLMs or backward the gradients through LLMs. Toolkens are appended at the last layer of the LLMs, making the cost similar to just doing inference. The authors have conducted extensive experiments and compared to proper baselines. The results are convincing.

**Strengths:**

The proposed method could make a good contribution to tool learning. In particular, the use of toolken is a novel yet elegant way to solve the massive tools adaption and generation problem.
The proposed method is efficient, yet to learn and extend.
The paper is clearly written and easy to follow.
The evaluation is comprehensive and convincing.

**Weaknesses:**

It would make the evaluation stronger if the authors can include some of the following experiments:

1. Lacking comparison with fine-tuning-based methods, e.g., toolformer. In Numerical Reasoning tasks, I believe these methods could work quite well.

2. The target is the massive tool-learning scenario,  it would be great to try some tasks that involve multiple quite different tool calls at the same time, e.g., qa and calculator. Although the experiments on VirturalHome are great, these 58 tools are mostly similar.

3. Testing whether the training and inclusion of toolken will affect the language generation capacity, i.e., adding some NLU tasks to see if there is any performance drop.

**Questions:**

NA

---

> ### Author Rebuttal · Authors · 2023-08-10
>
> We thank reviewer Mpoh for acknowledging that our proposed method is novel and elegant to learn massive tools efficiently, and makes a good contribution to the field of tool learning. We address the reviewer's comments on the evaluation as follows:
>
> (a) Comparing with full-model fine-tuning. Thanks for the great comment! For the comparison with Toolformer, please see our global response (a).
>
> (b) **Consider different tools simultaneously**. Thanks for the great suggestion and we appreciate the interesting setting pointed out by the reviewer. We initially focused on demonstrating the potential of ToolkenGPT in a scenario where a set of task-specific tools are used to solve one single task. As the reviewer mentioned, we've tried our best to increase the number and the diversity of such tools, i.e., 13 tools in math, 234 tools in KBQA, and 58 tools in VirtualHome. Since these tools are created for similar goals in specific tasks, they somehow are related to each other due to the nature of the evaluated tasks. Considering distinct tools, like QA and calculator, would require creating more challenging scenarios and tasks. We believe that such a set of experiments involving multiple, diverse tool calls could further highlight the flexibility and capability of ToolkenGPT, which serves as a realistic, multi-tool scenario for future works.
>
> (c) **Influence of Toolken on LLM generation**. Thanks for such an interesting comment! The potential influence on language generation is an interesting angle to discuss further. First of all, the training of toolken won't affect the generation capability of LLMs, as it doesn't change any pre-trained parameters for the generation. Second, for some NLG tasks like the summarization task that don't require external tools, the pure generation capacity from LLMs is enough, and thus, we can easily disable the learned tools by masking their toolkens in the output vocabulary, keeping the original generation capacity intact. This serves as a demonstration of the plug-and-play capacity of ToolkenGPT, as explained in the global response (a).
>
> Last but not least, in the most generic setting where we just keep the learned toolken embeddings and simply prompt the LLM for a generation task, we conducted an additional experiment to test the influence of toolken embeddings on NLG tasks. Specifically, we leverage the trained toolkens from math datasets and apply ToolkenGPT to two data-to-text generation datasets, WebNLG [1] and E2E [2]. We performed 10-shot learning (_randomly picked_) on the dev set of the WebNLG dataset (872) and the test set of the E2E dataset (630). We report the traditional NLG metrics, ROUGE-2 score from Huggingface evaluate and use greedy decoding for the generation.
>
> RTable 3: Performance of ToolkenGPT on NLG tasks.
>
> | Dataset      | Llama-30B | Llama-30B w/ toolkens | \#Func Calls |
> |--------------|----------|-----------------------|------------|
> | WebNLG      | 0.56     | 0.56                  | 0          |
> | E2E       | 0.46     | 0.46                  | 0          |
>
> RTable 3 shows that the inclusion of toolkens does not have any influence on the generation capabilities of LLMs on these two NLG tasks. As we can see from RTable 3, toolkens are not activated during the generation with the number of function calls as zero. However, in rare cases where the generation tasks are very similar to the toolken training text, the LLM may mistakenly call external tools, which is an interesting problem to investigate in the future. Nonetheless, we hope these additional results could further clarify the impact of toolken embeddings on NLG tasks and we will include them in the revised version.
>
> **References**:
>
> [1] Claire Gardent, Anastasia Shimorina, Shashi Narayan, and Laura Perez-Beltrachini. The WebNLG challenge: Generating text from RDF data. ICNL, 2017.
>
> [2] Jekaterina Novikova, Ondřej Dušek, and Verena Rieser. The E2E dataset: New challenges for end-to-end generation. arXiv preprint arXiv:1706.09254, 2017.

---

### Official Review · Reviewer_xBHJ · 2023-07-06

**Soundness:** 4 excellent
**Presentation:** 3 good
**Contribution:** 3 good
**Rating:** 8
**Confidence:** 4

**Summary:**

The paper presents ToolkenGPT, a framework for extending LMs with tool use. For each new tool, a new token is added to the output vocabulary of the LM and the embedding for that token is trained with annotated or synthetic examples. When the LM generates that token, the model is switched to a different mode and prompted with examples for that particular tool in order to generate the necessary arguments for that tool call.

**Strengths:**

The method seems novel and useful, pushing forward the area of research on extending LM capabilities by teaching them to call external tools.

**Weaknesses:**

Evaluation on three different types of tasks is great. However, the choice of baselines is the main weakness of the paper. The method seems to use substantial amounts of data for fine-tuning the token embeddings, whereas all the chosen baselines are only restricted to zero-shot or few-shot prompting. Comparison with Toolformer, which fine-tunes the whole model using similar data, would be more fair.

There are a number of writing errors. Please make sure to proof-read the paper.

**Questions:**

On line 152, "and does requires the gradients of LLM parameters" should presumably be "and does not require the gradients of LLM parameters"?

When generating the training data, do you do any filtering to make sure that the tool can actually correctly handle the given query? Seems like that would help.

For most of the tasks it is unclear how many synthetic data examples you generate. Please clarify.

With in-context learning and 234 tools to describe, are you actually able to fit all the tool examples or descriptions into the context? Or does some of this information simply get truncated? Please clarify.

**Limitations:**

The potential risks of letting LMs make calls to external APIs, with automatically generated arguments, should also be addressed.

---

> ### Author Rebuttal · Authors · 2023-08-10
>
> We thank the Reviewer xBHJ for acknowledging that our method is novel and useful, and will push forward the field of tool-augmented LLMs. We appreciate the reviewer's great suggestions and questions. We will further proofread the paper and fix the typos in the final version. Herein, we address the reviewer's questions as follows:
>
> (a) **Choices of baselines**. Thanks for the thoughtful comment! We thank the reviewer for recognizing the coverage of our evaluation and please refer to our global response (a) for following the reviewer's suggestion to compare with Toolformer and fine-tuning a model on similar data.
>
> (b) **Training data generation and filtering**. Thanks for the great clarification question and we apologize if the process of generating training data is not very clear in the experiment section. As shown in the appendix, for math datasets, we prompt ChatGPT to produce questions and their corresponding answers. Instead of giving specific numbers, we instruct the model to use placeholders that follow our particular format. We retain only those cases that are parsable and align with our format requirements. Afterward, we programmatically assign random numbers and compute the results.
>
> For example,
> ```
> Question: If the price of a stock increases by 10% every day, by how many times will its initial value increase after [ARG_0] days?
> Answer: After [ARG_0] days, the value of the stock will increase by a factor of 1.1^[ARG_0]=<power>(1.1, [ARG_0])=[ANSWER]
> Args: ['int']
> ```
> (c) **Number of synthetic data examples**. Regarding the training samples we synthesized, for the math dataset FuncQA, we synthesize 50 examples for each operation; for KAMEL, we synthesize at most 50 examples per relation (some examples are not successfully parsed and removed from the dataset, leading to 40 examples on average per relation).
>
> (d) **In-context learning on 234 tools**. Thanks for the question! It would be infeasible to fit all tool examples into the context. As indicated in Section 4.2 and Figure 2, we tried our best to fit as many tools into the context, reaching the 2048-tokens limit of LLaMA, which is about . The detailed mechanism of this baseline will be clarified in the revised version.
>
> (e) **Discussing potential risks of connecting LLMs with external APIs**. We appreciate this great comment on the potential risks of tool-augmented LLMs. We believe addressing such a potential risk requires community-wide efforts. We will delve deeper into this issue in the future work section of our revised paper.

---

> > ### Comment · Reviewer_xBHJ · 2023-08-21
> >
> > Thank you for your response. I have reviewed the reply and stick with my original score, which is already quite high.
> >
> > I think the inclusion of RTable 1 is useful - as this is a parameter-efficient method, it is expected that it doesn't perform quite the same as some more resource-hungry approaches.
> >
> > However, RTable 2 doesn't seem particularly useful. It just shows that Toolformer was trained for longer using more powerful GPUs - something that could be easily done with ToolkenGPT as well. Without knowing the performance, or at least training time, for both systems, there is no real comparison point.

---

### Official Review · Reviewer_C7TV · 2023-07-06

**Soundness:** 3 good
**Presentation:** 4 excellent
**Contribution:** 3 good
**Rating:** 7
**Confidence:** 4

**Summary:**

This study introduces an approach to teach Large Language Models (LLMs) to utilize numerous tools through learning embeddings for these tools. The embeddings are incorporated into the LLMs' output layer, invoking the use of tools at relevant steps. The inference process alternates between reasoning and tool-use modes to tackle novel tasks. Empirical results validate the efficacy of ToolkenGPT in utilizing a massive number of tools across various tasks, including numerical reasoning, knowledge-based question answering, and embodied plan generation.

**Strengths:**

- The proposed method is innovative and scales well to a massive number of tools with limited examples.
- The method is effective across a variety of tasks and demonstrates advanced multi-turn planning capability.
- The paper is well-structured and clearly written.

**Weaknesses:**

- While the authors assert the training efficiency of ToolkenGPT, empirical results reflecting computation costs and performance comparisons are missing.
- Given the current feasibility of instruction tuning (e.g., Alpaca, Vicuna), it would be beneficial to compare the proposed method with the fine-tuning of the entire model using the same training data. This comparison could clarify if there's a tradeoff between performance and computation cost.
- Although the authors claim a plug-and-play capability for ToolkenGPT, this feature hasn't been evaluated in the empirical studies. Additionally, if I understand correctly, we still need to train the embeddings for new tools. The authors should clarify this plug-and-play capability in more detail.

**Questions:**

- Is it possible to perform instruction tuning using the same training data?
- Can ToolkenGPT facilitate the zero-shot plug-and-play utilization of new tools?

**Limitations:**

The authors have discussed the limitations thoroughly.

---

> ### Author Rebuttal · Authors · 2023-08-10
>
> We thank the reviewer C7TV for acknowledging that our method is innovative, effective and scales well to massive tools with limited examples. Below we address the reviewer's comments and questions point by point.
>
> (a) Computation costs and performance. Thanks for the great comment! We understand the importance of demonstrating computational efficiency and we present additional results on computation resources and performance in global response (a) and we will include such results in the revised version.
>
> (b) Comparing with full-model fine-tuning. Thanks so much for the great suggestion! Please refer to the global response (a) to see a direct comparison with full-model fine-tuning.
>
> \(c\) Clarifying Plug-and-play capability. Thanks for this great clarification question! We made an explanation of the term plug-and-play in the global response, and we will update the paper in the final version to make it clearer.
>
> (d) Feasibility of instruction tuning. We appreciate this great suggestion! Based on our understanding, instruction tuning for tools shares a great overlapping with full-model fine-tuning using the same training data, as explained in the previous response (b). Some minor revisions might include mixing multi-tool data together and changing the prompt format a little bit. Therefore, we believe instruction tuning could be feasible with the same training data, although the computation cost may vary depending on the training methods for the instruction tuning. Further investigation into this direction of tuning the full model on a collection of API instructions is a very promising direction for the future research of tool-augmented LLMs.
>
> (e) Zero-shot plug-and-play for new tools. Thanks for the great suggestion! Please refer to our general response (b) for discussing the "plug-and-play" capacity of ToolkenGPT.

---

> > ### Comment · Reviewer_C7TV · 2023-08-18
> >
> > Thank you for providing the detailed responses. My concerns are well addressed.

---

### Author Rebuttal · Authors · 2023-08-10

We are grateful to all reviewers for their detailed and constructive feedback! We are encouraged to see that reviewers find:
- Our proposed ToolkenGPT is an "innovative, novel and useful", "a novel yet elegant way" (R-C7TV, R-xBHJ, R-QLCo, R-Mpoh), which "brings up new ideas", "fosters more research" and "pushes forward the area of research" on tool-augmented LLMs (R-xBHJ, R-5YmU, R-QLCo);
- ToolkenGPT is "efficient, yet to learn and extend" (R-Mpoh), offers a "much more efficient" method (R-5YmU) that "scales well to a massive number of tools with limited examples" (R-C7TV);
- Our "experimental setup is sound" with "well-designed baselines" (R-QLCo), our "empirical results on multiple domains" verify its advantage (R-5YmU), "the evaluation is comprehensive and convincing" (R-Mpoh)
- ToolkenGPT "is easy to implement and is able to be applied for various domains" (R-5YmU), and we "provide enough information for reproducing the experiments" (R-QLCo)

We have addressed all the questions raised by reviewers with additional experiments and thorough clarifications via separate responses to each reviewer. There are two common concerns we would like to address in the global response.

(a) **Comparison with full-model fine-tuning**. As shown in Table 1 from the paper, there are many advantages of ToolkenGPT in comparison with full-model fine-tuning, including training efficiency, quick adaptation to new tools, scaling to massive tools with limited examples, etc. While most reviewers agree on the above advantages, it is also interesting to compare with the full-modal fine-tuning, in terms of computation efficiency and performance. We implement different methods with Llama-7B on FuncQA, and show the comparison in RTable 1.

RTable 1: Comparison between ToolkenGPT and finetuning (using LoRA) in terms of training cost and performance on FuncQA dataset (using Llama-7B).

| Model           | One-hop  | Multi-hop | Computing Resource  | Training Time (all) |
|-----------------|---------|----------| ---|---|
| Fine-tune w/ LoRA |   0.62 |     0.074 | 1 \* A100 (80G) | ~40 min |
| ToolkenGPT |   0.55  |     0.058 |  1 \* RTX3090 (24G) | ~ 2 min |
| ReAct | 0.40   |  0.030  | - | -  |
| Baseline | 0.10  |  0.00    | - | - |

Though finetuning performs better than ToolkenGPT, it's significantly more expensive. Even though we use LoRA (parameter-efficient finetuning) and more computing resource (e.g., GPU memories, to increase the batch size), finetuning still costs about 20× more time than training toolken embeddings.

We also compare ToolkenGPT and Toolformer [1], a recent full-model fine-tuning method, in terms of efficiency on the learning of mathematic tools. Note that it's infeasible for us to compare performance, as Toolformer is not open-sourced and is based on GPT-J, a different base LM than ours (Llama). It's also computationally infeasible for us to reproduce Toolformer, given the extreme computational cost as shown in the table below.

RTable 2: Comparison between ToolkenGPT and Toolformer (math).

| Method | Computing Resource | Number of Training Examples |Time (per epoch)|
|---|---|---|---|
| ToolkenGPT (30B) | 4 \* RTX3090 (24GB) | 5k | ~16min |
| ToolkenGPT (13B) | 2 \* RTX3090 (24GB)| 5k | ~7min |
| ToolkenGPT (7B) | 1 \* RTX3090 (24GB)| 5k | ~5min |
| ToolFormer | 8 \* A100 (40GB) | 25k (only for math) | Unknown |

RTable 2 presents the comparison of computing resources and time for training our ToolkenGPT and Toolformer. It's clearly evident that our ToolkenGPT can be trained with much less computing resources, compared with the full-model fine-tuning method, Toolformer. Note that Toolformer's training time is not reported in the original paper.

(b) **Plug-and-play capacity of ToolkenGPT**. As mentioned by R-C7TV and R-QLCo, the definition of "plug-and-play" in Table 1 of the paper could be further clarified. By “plug-and-play”, we mean that new toolken embeddings are disentangled with other parameters, and can be enabled/disabled or directly added to an existing tool set effortlessly. The "plug-and-play" defined in the paper doesn't mean there is no need to train the embedding, but refers to that we are able to reuse the trained toolken embeddings and plug them into any downstream settings flexibly when needed. We will clarify this term in the updated paper.

Besides, it's also possible to train toolken embeddings without preparing training data manually. As demonstrated in Section 4.1 (FuncQA) and 4.2 (ToolkenGPT-syn), we can synthesize the training data for new tools. We can observe that ToolkenGPT (syn)---no in-domain training data is used---outperforms strong CoT baselines.

**References**

[1] Schick, Timo, Jane Dwivedi-Yu, Roberto Dessì, Roberta Raileanu, Maria Lomeli, Luke Zettlemoyer, Nicola Cancedda, and Thomas Scialom. "Toolformer: Language models can teach themselves to use tools." arXiv preprint arXiv:2302.04761 (2023).

[2] Miao, Shen-Yun, Chao-Chun Liang, and Keh-Yih Su. "A Diverse Corpus for Evaluating and Developing English Math Word Problem Solvers." In Proceedings of the 58th Annual Meeting of the Association for Computational Linguistics, pp. 975-984. 2020.

---

> ### Author Response · Authors · 2023-08-11
> **Correction: Computing resource for finetuning (LoRA)**
>
> There is a typo in the RTable 1: finetuning (LoRA) indeed used 8 x A100 (80G), instead of 1 x A100. So it costs much more computing resources than ToolkenGPT.

---

### Decision · Program_Chairs · 2023-09-21

**Decision:**

Accept (oral)

**Comment:**

The authors propose ToolkenGPT, an approach to teach language models to use tools that requires no modifications of the underlying parameters of the LLM. Essentially, separate embeddings (tool tokens -> toolkens) are learned in the output layer of  the LLM. These separate embeddings, when used in generation, invoke tool use. The reviewers have a consensus that this paper is innovative, scales well for use with many tools, and they like the efficiency and modularity of the approach. There were some concerns about how the approach fares against fine-tuning, which was addressed by the authors. Essentially, the model outperforms zero-shot and few-shot baselines handily, but is not quite as performant as fine-tuning. This makes sense, given that the underlying LLM parameters are frozen. However, the benefits of ToolkenGPT are that it can be easily extended to new tools, is much cheaper to adapt (since underlying LLM parameters can remain fixed, although their gradients do need to be estimated to update the too lembeddings), and moreover, does not harm the generation quality of the LLM through catastrophic forgetting. All reviewers agree that the paper makes a solid contribution to the field, and we hope that the reviews and discussions during the rebuttal paper make it into the final version of the paper (i.e. more detailed discussion of fine-tuning, terminology change of “massive tools” -> “massive amounts of tools”, clarifications about what is meant by “plug-and-play”, etc.).